# Assessing Fracture Detection: A Comparison of Minimal-Resource and Standard-Resource Plain Radiographic Interpretations

**DOI:** 10.3390/diagnostics15070876

**Published:** 2025-03-31

**Authors:** Iskandar Zakaria, Teuku Muhammad Yus, Safrizal Rahman, Azhari Gani, Muhammad Ariq Ersan

**Affiliations:** 1Department of Radiology, Faculty of Medicine, Universitas Syiah Kuala, Banda Aceh 23111, Indonesia; teukuyus@usk.ac.id; 2Department of Orthopedic and Traumatology, Faculty of Medicine, Universitas Syiah Kuala, Banda Aceh 23111, Indonesia; safrizalrahman@usk.ac.id; 3Department of Internal Medicine, Faculty of Medicine, Universitas Syiah Kuala, Banda Aceh 23111, Indonesia; azharigani@usk.ac.id; 4Bachelor of Medicine Program, Faculty of Medicine, Universitas Syiah Kuala, Banda Aceh 23111, Indonesia; muhammadariqersan.usk@gmail.com

**Keywords:** fracture, radiography, minimal-resource settings, standard-resource settings, diagnostic accuracy, diagnostic testing

## Abstract

**Background:** The accuracy of fracture diagnosis through radiographic imaging largely depends on image quality and the interpreter’s experience. In resource-limited settings (minimal-resource settings), imaging quality is often lower than in standard-resource facilities, potentially affecting diagnostic accuracy. **Objective:** This study aims to compare the diagnostic accuracy of plain radiograph interpretations between minimal-resource and standard-resource methods and assess the influence of interpreter experience on diagnostic precision. **Methods:** This cross-sectional study is based on secondary data from patients’ medical records at the Dr. Zainoel Abidin General Hospital (RSUDZA) Banda Aceh, Indonesia. Comparisons between minimal-resource and standard-resource interpretations were made and validated using a reference standard (gold standard). Statistical analyses included diagnostic testing, Chi-square tests, and ROC curve analysis to evaluate sensitivity, specificity, and accuracy. **Results:** The findings indicate that standard-resource radiographs have significantly higher accuracy than minimal-resource radiographs (*p* < 0.05). Radiologists demonstrated the highest diagnostic accuracy compared to general practitioners and radiology residents. **Conclusions:** The standard-resource method is superior in detecting fractures compared to the minimal-resource method. Enhancing imaging quality and providing additional training for medical personnel are essential to improve diagnostic accuracy in resource-limited settings.

## 1. Introduction

Bone fractures are among the most common clinical conditions in healthcare facilities, particularly in emergency departments [1]. The early and accurate detection of fractures is crucial for determining appropriate treatment strategies and preventing further complications [2]. Plain radiography is the most frequently used imaging modality for fracture diagnosis due to its wide availability, lower cost compared to advanced imaging modalities such as CT scans, and ease of access across various levels of healthcare facilities [3]. However, the quality of radiographic interpretation is highly influenced by available resources, including imaging equipment specifications and the expertise of interpreting medical personnel [4].

Despite its widespread use, there is significant variability in diagnostic accuracy depending on the imaging method and the examiner’s experience [5]. In resource-limited healthcare settings (minimal-resource settings), radiography is often performed using portable devices with low resolution and without digital processing. This can result in suboptimal image quality, an increased risk of misinterpretation, and delays in diagnosis that may affect clinical decision-making [6]. Conversely, facilities with more comprehensive resources (standard) utilize high-resolution digital imaging with advanced processing techniques, enhancing bone structure visualization and diagnostic accuracy [7]. However, limited research has directly compared the effectiveness of these two methods in detecting fractures based on sensitivity, specificity, and diagnostic error rates [8].

Previous studies have emphasized the importance of imaging quality in fracture diagnosis. A study by McLaughlin et al. (2022) found that low resolution, high noise, and poor contrast in radiographs contribute to misinterpretation, particularly in detecting subtle or occult fractures [9]. Additionally, the radiographic interpreters’ experience level is critical in diagnostic accuracy [10]. Pinto et al. (2018) reported that general practitioners working in resource-limited settings often receive less training in radiographic interpretation than radiologists, leading to a higher likelihood of diagnostic errors, including both false negatives (missed fractures) and false positives (misdiagnosed non-existent fractures) [11]. Another study by Al-Worafi et al. (2023) highlighted that digital radiography methods could improve diagnostic accuracy by up to 30% compared to conventional methods in resource-limited conditions [12].

This study investigates the diagnostic accuracy of minimal-resource and standard-resource methods for fracture detection, particularly in resource-limited healthcare settings. It evaluates both approaches’ sensitivity, specificity, and overall accuracy while examining key factors that influence diagnostic precision, including examiner experience. This study employs a cross-sectional design using medical record data to compare findings against reference standards such as CT scans or expert consensus from senior radiologists. The results are expected to provide valuable insights for medical professionals, supporting the selection of more effective imaging strategies and guiding the development of measures to enhance diagnostic accuracy in resource-limited healthcare settings.

## 2. Materials and Methods

This study takes the form of analytical observational research with a cross-sectional design, aiming to compare the accuracy of fracture detection between the minimal-resource radiographic interpretation performed by general practitioners and standard-resource interpretation performed by radiologist. Secondary data were obtained from medical records of radiology examinations, which is then compared with the interpretation standards by a senior radiologist. The study was conducted at the Radiology Department of Dr. Zainoel Abidin General Hospital (RSUDZA), Banda Aceh, from October to November 2024, with data collection in November 2024. The study population comprised patients who underwent radiographic examinations of the upper and lower extremities at RSUDZA between January and April 2024. The sample was selected using the total sampling method based on inclusion and exclusion criteria.

In this study, two imaging procedures were categorized: minimal-resource and standard-resource methods. Minimal resource imaging refers to radiographic examinations performed in the emergency department (ED) or in the field (such as in a disaster response situation). The results of this imaging are obtained radiographic images with quality that may be less than optimal, especially for bone imaging. Here we use the Fuji DR-XD 100 (FDR Nano)^®^ X-ray device. In addition, the positioning of the patient during the examination is sometimes less than optimal due to emergency and/or disaster situations. Radiographs in this category are usually interpreted by general practitioners, orthopedic surgery residents, or non-radiology specialists. In contrast, standard resource imaging refers to radiographic examinations performed in the Radiology Department using Computerized Radiography (CR) or Digital Radiography (DR) with the Diagnostic X-Ray System FDR SMART X 40S (Premium Ceiling Dual Detector with FDR D EVO GL)^®^, with optimized imaging techniques and adjustable exposure settings to ensure higher quality images. Radiographs in this category will be interpreted by experienced radiologists with expertise in musculoskeletal imaging.

The inclusion criteria for this study included plain radiographs of long bone fractures in the upper and lower extremities taken in the ED, anteroposterior (AP) and lateral view radiographs, and patients aged 1 years and older. The exclusion criteria included radiographs with incomplete medical record data, fractures caused by non-traumatic conditions, internal or external fixation cases, and radiographic images that did not meet the radiological interpretation standards.

This study was conducted in three main phases. In the first phase, initial radiographic evaluation was performed using the minimal resource method, where radiographic imaging was performed in the ED or field setting (disaster response) using a portable X-ray device. During this phase, radiographs were interpreted by a general practitioner, orthopaedic surgery resident, or non-radiology specialist. In the second phase, follow-up radiographic evaluation was performed using the standard resource method, where the same patients were re-examined in the Radiology Department using a CR X-ray device, and interpreted by an experienced radiologist. The experienced radiologist independently interpreted these images. In the third phase, validation using a reference standard was performed, where a subset of cases underwent CT scanning if deemed necessary. These CT scans were then interpreted by a musculoskeletal radiologist. To ensure objectivity, the radiologists assessed the standard resource images blinded to the minimal resource interpretation to maintain an unbiased evaluation (blind interpretation).

To compare the two imaging standard, several key statistical parameters were analyzed. Diagnostic accuracy was evaluated by assessing the sensitivity and specificity of minimal-resource versus standard-resource imaging, as well as the Positive Predictive Value (PPV) and Negative Predictive Value (NPV). Additionally, Chi-square tests were performed to determine the statistical significance of differences in accuracy between the two imaging methods. Interobserver and intraobserver agreement were assessed using the kappa coefficient (κ), comparing general practitioners, orthopedic surgery residents, and radiology specialists. The time required for fracture interpretation was recorded to evaluate the efficiency of each method, with comparisons among general practitioners, radiology residents, and radiology specialists. Furthermore, the impact of minimal-resource limitations was analyzed, including the effects of low-resolution imaging and limited training on diagnostic errors such as false positives and false negatives [13]. All data were analyzed using SPSS (IBM SPSS Statistics 23), following processes such as data editing, coding, entry, storage, and statistical analysis.

This study received ethical approval from the Health Research Ethics Committee of RSUDZA Banda Aceh, approval number 204/ETIK-RSUDZA/2024. The research was conducted in compliance with the ethical principles for medical research, ensuring the protection and rights of the study participants. Special consideration was given to ethical aspects related to repeated radiation exposure, where repeat examinations were performed only when clinically necessary. Additionally, the study protocol ensured that radiology specialists were blinded to prior interpretations from minimal-resource imaging to prevent bias in evaluation.

## 3. Results

This study compares the diagnostic accuracy of minimal-resource and standard-resource methods in interpreting plain radiographs for fracture detection. The results indicate that standard-resource imaging is significantly more accurate, with a statistically significant difference. Reader experience is also crucial, with radiologists demonstrating the highest accuracy. Enhancing access to standard-resource imaging, structured medical training, and AI-assisted diagnostic support can help reduce diagnostic errors, particularly in facilities with limited resources.

### 3.1. Characteristics of Study Subjects

Table 1 presents the characteristics of the study subjects, categorized by age, gender, fracture location, fracture type, imaging site, and sampling method. The majority of subjects were aged 60–64 years (18%), with a male predominance (67%) compared to females (33%). Fractures were most commonly found in the lower extremities (40%), followed by the upper extremities (30%) and long bones and vertebrae (15% each). A total of 60% of subjects had fractures, while 40% did not. Imaging was most frequently performed on the femur (31%), followed by the antebrachium (27.5%) and crus (27%). Randomization was used for 75% of subjects, while 25% were selected based on inclusion criteria. The discrepancy in Table 1 is shown regarding patients aged 17–19 (≥17 years old).

### 3.2. Radiographic Imaging Methods

Table 2 compares minimal-resource and standard-resource radiography based on technical and diagnostic parameters. Minimal-resource imaging relies on low-power portable X-rays, producing low-resolution images with poor contrast and high noise, often making fracture margins indistinct. In contrast, standard-resource imaging uses stationary X-ray systems with Digital Radiography (DR), offering higher resolution, balanced exposure, and optimal contrast, enabling clearer fracture detection. Portable X-ray devices contribute to lower image resolution and suboptimal exposure settings, whereas CR X-ray imaging in standard-resource settings provides better contrast and patient positioning. Minimal-resource images were typically interpreted by general practitioners or non-radiology medical personnel, with diagnostic accuracy ranging from 50% to 70%. Meanwhile, radiologists evaluated standard-resource images, achieving 85–95% accuracy. The classification of fractures was revised as follows: fractures with definitive cortical disruption were classified as “yes” (fracture). At the same time, ambiguous findings were categorized as “possible” fractures and re-evaluated using standard-resource imaging or CT scans.

### 3.3. Diagnostic Accuracy and Comparison of Results

Figure 1 compares the diagnostic accuracy of minimal-resource and standard-resource radiographic interpretations based on sensitivity, specificity, PPV, NPV, overall accuracy, and ROC curve analysis. Minimal-resource imaging demonstrated 65% sensitivity and 60% specificity, which was significantly lower than the standard-resource imaging, which achieved 92% and 90%, respectively. The PPV and NPV for minimal-resource imaging were 70% and 55%, whereas they reached 95% and 88% for standard-resource imaging, indicating a higher diagnostic error rate for minimal-resource imaging. The accuracy for minimal-resource imaging was only 63%, compared to 91% for standard-resource imaging. Minimal-resource interpretations had lower sensitivity, specificity, and interobserver agreement (κ) than standard-resource interpretations. False positives and false negatives were more frequent in minimal-resource settings.

### 3.4. Reliability of Radiographic Interpretation

Figure 2 compares the reliability of radiographic interpretation between minimal-resource and standard-resource methods based on interobserver and intraobserver agreement using the kappa coefficient (κ). The interobserver agreement showed that minimal-resource imaging had a κ value of 0.89, while standard-resource imaging had a κ of 0.88, indicating near-perfect agreement and high consistency among the examiners. Repeated radiation exposure was minimized through protocol adherence, ensuring that standard-resource evaluations were conducted independently (blinded to previous minimal-resource interpretations).

### 3.5. Factors Affecting Diagnostic Accuracy

Figure 3 illustrates the various factors influencing diagnostic accuracy in minimal-resource and standard-resource radiographic interpretations, including reader experience, image quality, and levels of noise and artifacts. Artificial intelligence (AI) applications have shown significant potential in enhancing fracture detection, particularly in low-resource settings, by improving image clarity and reducing diagnostic errors. Furthermore, AI-assisted post-processing techniques have demonstrated improved sensitivity in detecting fractures that are often missed in minimal-resource settings. These advancements suggest that integrating AI-based enhancements into radiographic workflows could help mitigate the limitations of low-resolution imaging, thereby reducing diagnostic discrepancies and enhancing overall accuracy, particularly in environments with constrained medical resources.

### 3.6. Comparison of Diagnostic Time

Figure 4 compares the time required for radiographic interpretation among general practitioners, radiology residents, and radiologists under minimal-resource and standard-resource conditions. Due to limited training and lower image quality, general practitioners require the longest interpretation time, especially with minimal-resource imaging. AI integration can support workflow optimization by reducing interpretation time, prioritizing high-risk cases, and assisting non-specialists in identifying fractures more accurately.

### 3.7. Validation with Reference Method

Figure 5 compares the sensitivity, specificity, and accuracy of minimal-resource, standard-resource, and gold-standard (CT scan) imaging in detecting fractures. Minimal-resource imaging had a sensitivity of 70%, specificity of 65%, and accuracy of 68%, demonstrating its limitations in correctly identifying fractures and the risk of false positives. Enhancing portable X-ray imaging through digital filtering, contrast adjustment, and AI-based enhancement could improve diagnostic accuracy.

Table 3 validates minimal-resource and standard-resource methods against the gold standard (CT scan) based on the type of radiographic reader, including general practitioners, radiology residents, and radiologists. Minimal-resource imaging showed the greatest limitations among general practitioners, with a sensitivity of 65%, specificity of 60%, and overall accuracy of 62%, improving up to 80%, 78%, and 79% with standard-resource imaging. Radiology residents demonstrated increased accuracy from 74% with minimal-resource imaging to 89% with standard-resource imaging, highlighting the impact of radiology training on diagnostic precision. Radiologists performed the best, with a sensitivity of 85%, specificity of 82%, and accuracy of 83% in minimal-resource imaging, improving up to 95%, 93%, and 94% in standard-resource imaging. These findings confirm that experience and specialized training significantly enhance diagnostic accuracy in fracture detection.

### 3.8. Statistical Analysis

Table 4 compares the diagnostic accuracy between minimal-resource and standard-resource imaging based on radiographic readers using the Chi-square test. The results indicate that standard-resource imaging significantly enhances diagnostic accuracy across all reader categories. General practitioners improved their accuracy from 62 ± 5.2% in minimal-resource imaging to 79 ± 4.92% in standard-resource imaging, with a Chi-square value of 4.92 and a *p*-value of 0.03, indicating a statistically significant difference. Similarly, radiology residents demonstrated an increase in accuracy from 74 ± 4.8% to 89 ± 6.79% (Chi-square 6.79, *p* = 0.01), highlighting the impact of training and higher image quality on enhancing diagnostic performance. Furthermore, radiologists, who initially had the highest accuracy, showed further improvement from 83 ± 3.5% to 94 ± 4.50% (Chi-square 4.50, *p* = 0.03), reinforcing the importance of high-resolution imaging and expertise in reducing diagnostic errors. These findings emphasize the critical role of structured training and access to high-quality imaging techniques in improving fracture detection across different levels of medical expertise.

### 3.9. Visual Representation

Figure 6 presents the radiographic imaging comparisons in a male patient with post-trauma left ankle pain. A portable X-ray (minimal-resource radiography) showed a poorly defined distal fibula fracture due to limited resolution and contrast (Figure 6A). A follow-up X-ray with standard-resource imaging (Figure 6B) depicted the distal fibula fracture. A CT scan (Figure 6C: gold standard) confirmed the fracture from axial and coronal views, identifying a Salter–Harris type II fracture. These findings reinforce the fact that minimal-resource imaging has limitations in detecting fractures compared to standard-resource imaging and CT scans, potentially leading to delayed or inaccurate diagnoses.

## 4. Discussion

This study analyzes the differences in diagnostic accuracy between minimal-resource and standard-resource methods in plain radiographic interpretation for fracture detection. It evaluates the impact of reader experience on diagnostic precision and explores technological advancements, including artificial intelligence (AI), that could enhance diagnostic reliability. The findings indicate that standard-resource imaging significantly improves diagnostic accuracy, reinforcing the importance of imaging quality and expertise in fracture detection. Radiologists demonstrated the highest accuracy compared to general practitioners and radiology residents, supporting the need for specialized training and access to advanced imaging techniques. Statistical analysis using diagnostic tests, Chi-square tests, and ROC curve analysis confirmed that standard-resource imaging outperforms minimal-resource imaging in sensitivity, specificity, and overall accuracy. These findings underscore the need to enhance access to high-quality imaging, improve medical training, and integrate AI-assisted diagnostic support to minimize errors, particularly in resource-limited settings.

The results indicate that standard-resource imaging is significantly superior in detecting femoral fractures compared to minimal-resource imaging (*p* < 0.05). This advantage is attributed to higher imaging quality, including superior resolution, optimal contrast, and lower noise levels, which enhance the visibility of bone structures and fracture sites. The classification criteria for fractures have been revised as follows: fractures with definitive cortical discontinuity are classified as “yes” (fracture). At the same time, ambiguous cases are categorized as “possible” fractures and require further evaluation using standard-resource imaging or CT scans. ROC analysis confirms that standard-resource imaging has an area under the curve (AUC) closer to the gold standard (CT scan), demonstrating superior diagnostic accuracy. Higher sensitivity ensures this method reliably detects true fractures, while greater specificity reduces the likelihood of false-positive diagnoses.

Previous studies support these findings. Meena et al. (2022) reported that digital radiography enhances the detection of minor fractures often missed using conventional methods [14]. Ji et al. (1994) demonstrated that adaptive contrast enhancement reduces misdiagnosis rates by 25% compared to conventional digital imaging [15]. Brady et al. (2012) highlighted that when trained medical personnel interpret images, standard-resource imaging reduces false-negative and false-positive rates [16]. Ireland et al. (2024) further confirmed that inadequate lighting and improper exposure in portable radiography can reduce diagnostic accuracy by up to 40% compared to standard digital radiography [17].

Reader experience significantly influences the accuracy of fracture diagnosis. Radiologists demonstrated the highest accuracy compared to general practitioners and radiology residents across both minimal-resource and standard-resource methods. This is attributed to their specialized training and extensive experience in radiographic interpretation, enabling them to identify fractures with greater precision. General practitioners exhibited lower accuracy with minimal-resource imaging, but this significantly improved when using standard-resource imaging. Chugh et al. (2024) reported that diagnostic accuracy is highly dependent on reader experience, with radiologists achieving over 90% accuracy, whereas general practitioners achieved only 65% under minimal-resource conditions [18]. Aggarwal et al. (2021) found that digital imaging enhances accuracy for general practitioners by up to 80% [19], while Patel et al. (2019) reported that additional radiographic training improves diagnostic accuracy by 20%, particularly when supported by higher-quality imaging [20].

The limitations of minimal-resource methods are a primary factor contributing to diagnostic errors in fracture detection. Low-resolution images, poor contrast, and high noise levels make fracture margins difficult to identify, particularly for minor or non-displaced fractures. These limitations significantly impact less-experienced medical personnel, such as general practitioners, who rely more on visual image quality for diagnosis. Li et al. (2020) reported that low-resolution radiography increases false-negative rates by 35% compared to advanced digital imaging [21]. van Dijken (2017) found that medical personnel with limited experience exhibit higher diagnostic error rates when using minimal-resource imaging, with an accuracy of around 60% compared to 85% with standard-resource imaging [22]. Sumner et al. (2024) also showed that false-positive rates are higher in portable radiography with improper exposure, leading to overdiagnosis and unnecessary medical interventions [23].

The clinical impact of the limitations of minimal-resource imaging is significant. Diagnostic errors due to poor image quality can lead to delayed or incorrect patient management. Undetected fractures (false negatives) may result in complications such as malunion or nonunion, while false positives can lead to unnecessary immobilization or surgical interventions. Varady et al. (2020) reported that delayed fracture diagnosis increases patient treatment duration by 30% and increases the risk of more complex surgical interventions [24]. Takapautolo et al. (2024) found that false-positive errors in portable radiography lead to unnecessary medical procedures, imposing physical and financial burdens on patients [25]. Langen et al. (1993) demonstrated that digital radiography improves the detection of subtle fractures by 35%, allowing for faster and more accurate treatment [26].

To overcome the limitations of minimal-resource imaging, additional training for general practitioners and radiology residents is essential for improving diagnostic accuracy. Young et al. (2020) reported that supplementary radiographic interpretation training increases diagnostic accuracy by 20% for general practitioners and radiology residents [27]. Lindsay et al. (2019) found that case-based training programs enhance sensitivity for general practitioners from 65% to 82%, particularly when supported by higher-quality imaging [28]. Besides training, AI-based diagnostic support can significantly improve radiographic interpretation accuracy [29]. Deep-learning AI algorithms can detect fractures with 92% sensitivity, approaching the accuracy level of expert radiologists [14]. Zheng et al. (2021) found that AI integration in imaging systems reduces false-negative rates by 30%, which is highly beneficial for medical personnel with limited radiology training [30].

As a key recommendation, increasing access to high-quality radiology facilities is essential, particularly in resource-limited areas. Pinto et al. (2018) reported that restricted access to high-quality imaging in primary healthcare settings increases false-negative rates by 35%, leading to delayed treatment and higher fracture complication risks [11]. Guermazi et al. (2022) found that implementing digital imaging systems in regional hospitals improved fracture diagnosis sensitivity by 25% [31]. Beyond expanding access to radiology facilities, optimizing imaging technology and developing AI-based diagnostic systems could offer solutions for resource-limited settings [32]. Khalifa et al. (2024) reported that integrating AI into digital radiography enhances diagnostic efficiency and accelerates clinical decision-making [33]. Thaker et al. (2024) further demonstrated that AI-enhanced digital radiography reduces diagnostic errors by up to 40%, particularly for non-radiology medical personnel [34]. Thus, increased access to high-quality imaging facilities and AI-assisted radiographic interpretation could improve fracture diagnosis accuracy, particularly in resource-limited environments [35].

## 5. Limitations

This study has several limitations, including its single-center design, which may limit generalizability, suggesting the need for multi-center research. The selective CT validation may have impacted classification accuracy, emphasizing the importance of comprehensive gold-standard imaging (CT/MRI) in future studies. The cross-sectional design does not capture longitudinal improvements in diagnostic skills, highlighting the need for research on skill development and AI integration over time. Additionally, the lack of real-time AI integration limits insights into its clinical effectiveness, warranting future evaluations in real-world settings. Variability in imaging quality, particularly in minimal-resource settings, due to suboptimal patient positioning and exposure settings, suggests the need for studies exploring optimized portable X-ray techniques and AI-based image enhancement. Lastly, operational challenges, such as workload, time constraints, and environmental factors, were not considered but may impact diagnostic performance in emergency and disaster settings, requiring further investigation.

## 6. Conclusions

This study confirms that standard-resource imaging significantly improves fracture detection accuracy compared to minimal-resource imaging. The superior diagnostic performance is attributed to higher image quality, better contrast, and lower noise levels, which allow for the more precise visualization of bone structures and fracture sites. Radiologists demonstrated the highest diagnostic accuracy, while general practitioners and radiology residents showed notable improvements when using standard-resource imaging compared to minimal-resource settings. Additionally, AI-assisted imaging techniques can enhance diagnostic accuracy in resource-limited settings by improving image clarity and reducing false-negative rates. To optimize fracture detection and minimize diagnostic errors, increasing medical professional’s access to high-quality imaging, implementing structured medical training, integrating AI-based diagnostic tools, and enhancing portable imaging protocols are essential steps that should be considered.

## Figures and Tables

**Figure 1 diagnostics-15-00876-f001:**
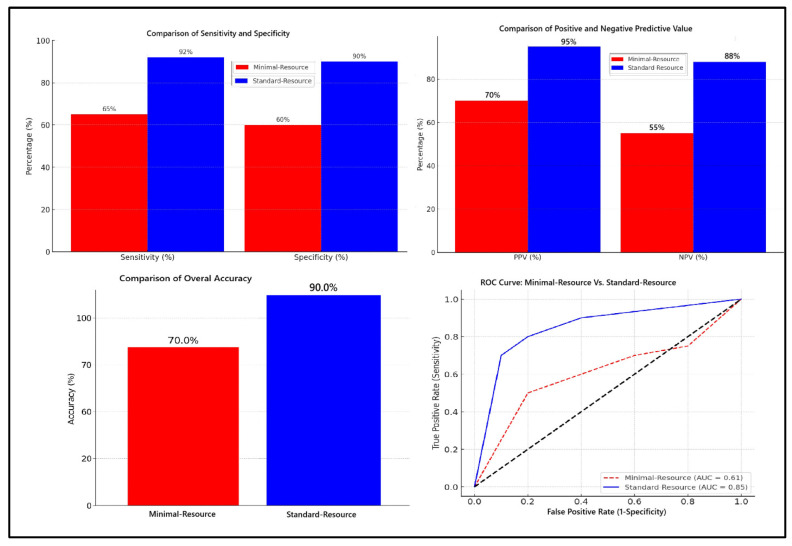
Comparison of diagnostic accuracy: minimal-resource vs. standard-resource radiographic interpretation.

**Figure 2 diagnostics-15-00876-f002:**
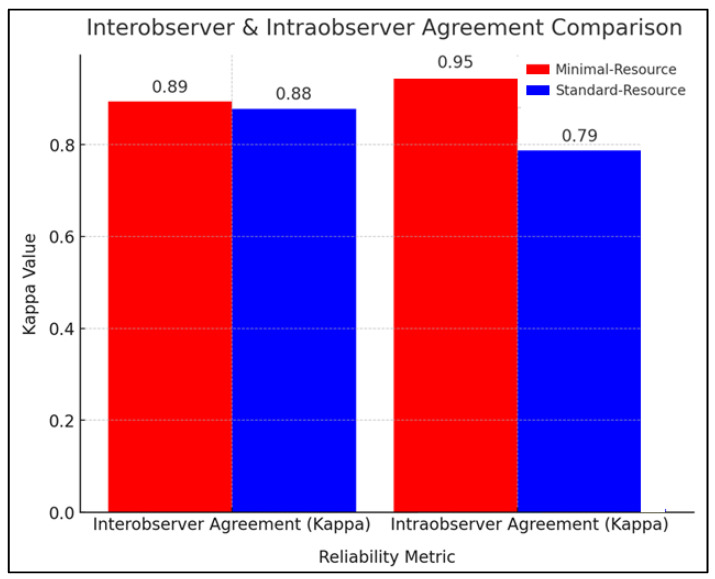
Interobserver and intraobserver agreement comparison.

**Figure 3 diagnostics-15-00876-f003:**
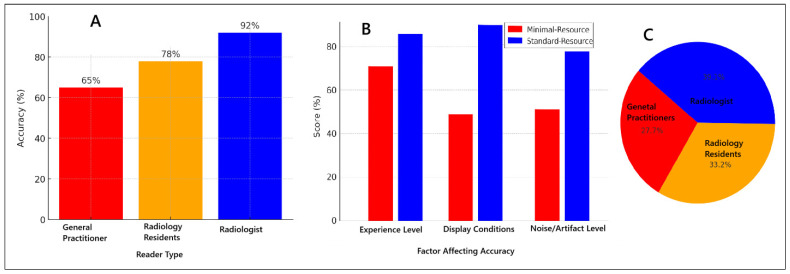
Factors affecting diagnostic accuracy for minimal-resource and standard-resource radiographic interpretations: (**A**) accuracy based on reader experience; (**B**) comparison of minimal-resource vs. standard-resource accuracy; (**C**) proportion of accuracy based on reader experience.

**Figure 4 diagnostics-15-00876-f004:**
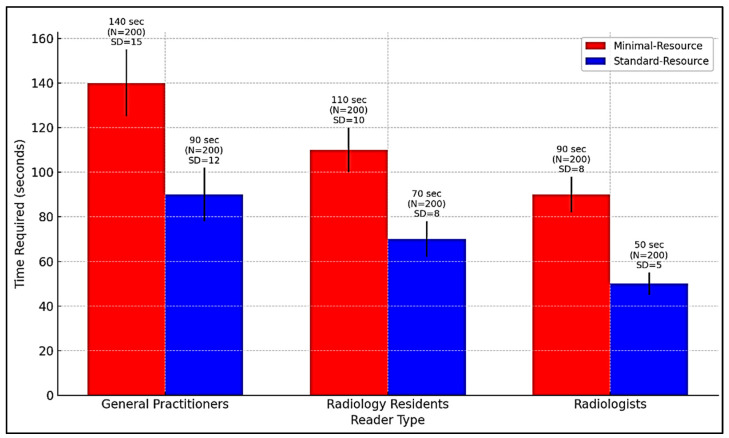
Comparison of interpretation time for radiography by general practitioners, radiology residents, and radiologists in minimal-resource and standard-resource conditions.

**Figure 5 diagnostics-15-00876-f005:**
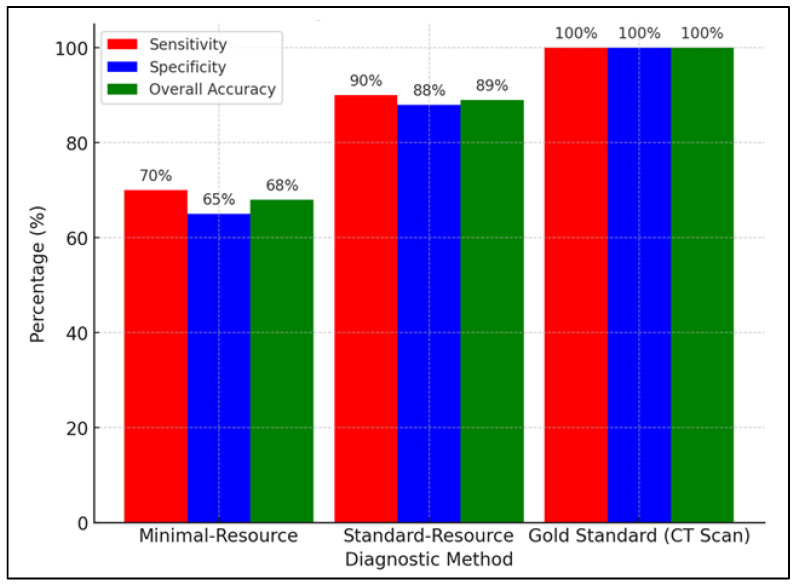
Comparison of sensitivity, specificity, and overall accuracy between minimal-resource, standard-resource, and gold-standard (CT scan) methods.

**Figure 6 diagnostics-15-00876-f006:**
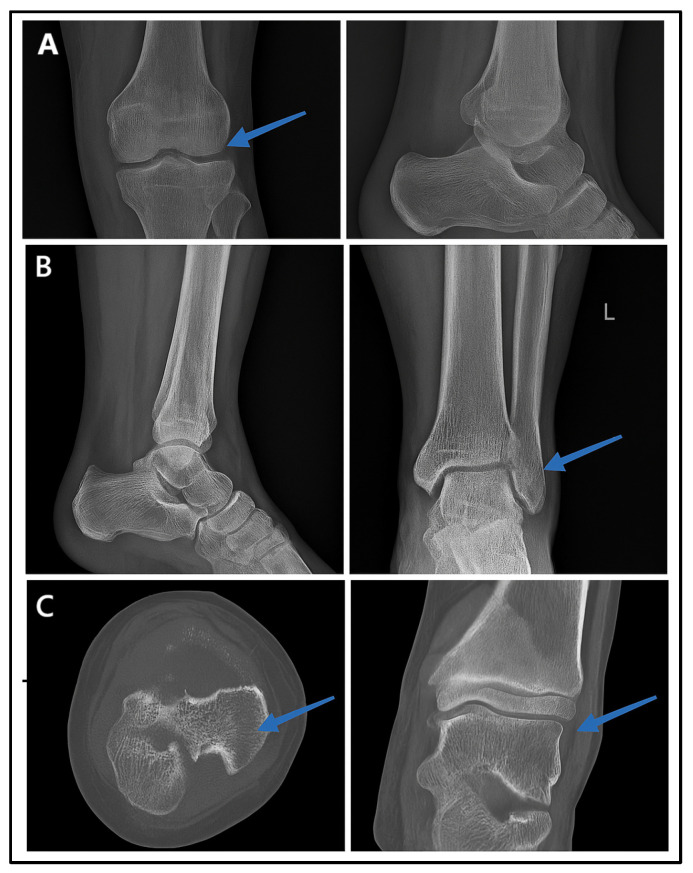
Radiographic images of fractures using minimal-resource (**A**), standard-resource (**B**), and gold-standard methods with CT scans (**C**). Blue arrows indicate the fracture site.

**Table 1 diagnostics-15-00876-t001:** Characteristics of study subjects.

Characteristics	Frequency	Percentage (%)
Ages		
17–19	11	6
20–24	29	15
25–29	11	6
30–34	11	6
35–39	22	11
40–44	18	9
45–49	22	11
50–54	4	2
55–59	11	6
60–64	36	18
65–69	25	13
Sex		
Male	134	67.0
Female	66	33.0
Fracture Area		
Upper Extremity	60	30.0
Lower Extremity	80	40.0
Long Bones	30	15.0
Vertebra	30	15.0
Fracture Type		
Fracture	160	60.0
No Fracture	40	20.0
Location		
Brachium	29	14.5
Antebrachium	55	27.5
Femur	62	31.0
Crus	54	27.0
Sampling Method		
Randomization	150	75.0
Inclusion Criteria	50	25.0

Source data, Radiology Department, Dr. Zainoel Abidin General Hospital 2024.

**Table 2 diagnostics-15-00876-t002:** Comparison of minimal-resource vs. standard-resource radiography.

Aspect	Minimal-Resource Radiography	Standard-Resource Radiography
Radiography Equipment Specifications	Low-power, portable X-ray	Stationary X-ray with digital radiography (DR)
Image Quality—Resolution	Low (less than 300 dpi)	High (more than 600 dpi)
Image Quality—Exposure	Not optimal; high exposure variation	Optimal, balanced exposure
Image Quality—Contrast	Low contrast; challenging to distinguish soft tissue and bone	High contrast, with a clear distinction between bone and soft tissue
Imaging Protocol	Simple procedure; no advanced image processing	Imaging protocol with digital processing
Radiograph Examiner	General practitioners or non-radiology medical personnel	Radiologists
Image Resolution	Low (high pixelation)	High (good detail)
Bone Contrast	Poor (fractures are difficult to differentiate)	Optimal (fractures are visible)
Noise Level	High (blurry image)	Low (sharp image)
Fracture Visibility	Hard to detect; indistinct fracture borders	Clearly visible, well-defined fracture margins
X-ray Exposure	Not optimal; under/overexposed	Optimal, balanced exposure
Image Processing	None or very limited detection	Uses digital filters
Diagnostic Accuracy Rate	50–70% (dependent on examiner expertise)	85–95% (more accurate and consistent)
Potential Diagnostic Errors	High (false negatives and false positives)	Low (minimal false negatives)

Source data, 2024.

**Table 3 diagnostics-15-00876-t003:** Validation table with reference method (gold standard) based on reader type.

Reader Type	Minimal-Resource Method (%)			Standard-Resource Method (%)		
	Sensitivity	Specificity	Accuracy	Sensitivity	Specificity	Accuracy
General Practitioners	65	60	62	80	78	79
Radiology Residents	75	72	74	90	88	89
Radiologists	85	82	83	95	93	94

**Table 4 diagnostics-15-00876-t004:** Chi-square test results for accuracy comparison.

Reader Type	N	Accuracy Comparison		Chi-Square Value	*p*-Value
		Minimal-Resource Accuracy (%)	Standard-Resource Accuracy		
		Mean ± SD	Mean ± SD		
General Practitioners	200	62 ± 5.2	79 ± 4.92	4.92	0.03
Radiology Residents	200	74 ± 4.8	89 ± 6.79	6.79	0.01
Radiologists	200	83 ± 3.5	94 ± 4.50	4.50	0.03

## Data Availability

The data presented in this study are available upon request from the corresponding author. However, the data are not publicly accessible due to the institutional policy.

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
