# Peer review of "Assessing Fracture Detection: A Comparison of Minimal-Resource and Standard-Resource Plain Radiographic Interpretations"

_diagnostics, 2025, doi:10.3390/diagnostics15070876_

Round 1

Reviewer 1 Report

Comments and Suggestions for Authors

This study aims to compare the diagnostic accuracy of minimal-resource versus standard-resource methods for fracture detection. The topic is relevant and interesting. The paper is well-structured, and the findings hold clinical significance, as fracture diagnosis should not rely solely on minimal resources. However, before publication, several issues need to be addressed.

In the materials and methods section, additional details are required regarding the radiology device models and their setup, as this would enhance reproducibility.

The methodology for fracture evaluation also needs further clarification. Was the same fracture assessed first with minimal resources, then with standard resources, and finally with a CT scan? This sequence is unclear from the current description. Additionally, how was the fracture interpreted? If a clear fracture site was identified, was it recorded as "yes"? How were probable fractures classified? These aspects need to be explicitly defined. What about the ethical aspects of repeated (and possibly unnecessary) radiological exposure?

In the results section, the first paragraph presents a conclusion and would be more appropriate in the discussion section.

In Table 1, study subject data includes patients aged 15–19 years, while the inclusion criteria specify patients over 17 years. This discrepancy should be clarified.

Table 2 should be moved to the materials and methods section, along with the paragraph from lines 138–149, as these details describe the methodology rather than the results.

Additionally, in Table 2, under “radiograph examiner,” minimal-resource assessments are listed as being performed by “general practitioners or non-radiology medical personnel,” while standard-resource assessments are attributed to “radiologists.” However, the results indicate that radiographs from both minimal and standard resources were evaluated by general practitioners, radiology residents, or radiologists. This inconsistency needs to be addressed to avoid confusion.

I suggest including this point in the limitations section: there are minor inconsistencies in fracture incidence between minimal-resource and standard-resource radiography, which could potentially influence the interpretation of results. Acknowledging this limitation would provide a more comprehensive perspective on the study's findings.

Respectfully submitted,

Author Response

Comment 1: Radiology Methodology Details: Additional details on radiology equipment and settings are needed to improve reproducibility.

Response 1: Details on radiology equipment will be added, including the differences between Minimal-Resource and Standard-Resource imaging. Minimal-resource imaging involves suboptimal imaging techniques, portable X-ray use, and non-radiology medical personnel. Standard-Resource imaging includes optimized imaging techniques, Computerized Radiography (CR) X-rays, and radiology specialists' evaluation.

Comment 2: Clarification of Fracture Evaluation Methodology: The sequence of fracture assessment is unclear—are fractures first assessed using minimal resources, then standard resources, and finally CT scans?

Response 2: The methodology section will clarify the evaluation sequence: Initial imaging is performed using Minimal-Resource methods in the ER or field settings, followed by repeat imaging in the Radiology Department using Standard-Resource methods for comparison. CT scans are performed in selected cases as needed and interpreted by radiology specialists.

Comment 3: Clarification of Fracture Interpretation: Further explanation is needed on how fractures are classified as "yes" and how possible fractures are determined.

Response 3: Fracture classification will be explained: A definite fracture is identified when clear cortical discontinuity is observed, categorized as "yes" for fracture. If findings are inconclusive, the case is labeled as a "possible fracture" and further evaluated using Standard-Resource imaging or CT scans.

Comment 4: Ethical Considerations of Repeated Radiation Exposure: The ethical aspects of repeated radiological exposure should be addressed.

Response 4: Ethical considerations will be included, particularly regarding repeated radiation exposure. The study protocol ensures that repeat imaging is conducted only when necessary, and radiologists evaluating Standard-Resource images are blinded to prior Minimal-Resource interpretations to avoid bias.

Comment 5: Data Structure and Presentation: The conclusion in the first paragraph of the results section should be moved to the discussion section.

Response 5: The conclusion in the first paragraph of the results section will be relocated to the discussion section

Comment 6: Data Structure and Presentation: Table 1 presents subjects aged 15–19 years, while the inclusion criteria state subjects must be over 17 years—clarification is needed.

Response 6: The discrepancy in subject age in Table 1 will be addressed by revising the age limits or adjusting the analyzed subject data to align with the inclusion criteria.

Comment 7: Data Structure and Presentation: Table 2 would be more appropriate in the materials and methods section as it describes methodology.

Response 7: Table 2 will be moved to the materials and methods section to ensure structural consistency.

Comment 8: Data Structure and Presentation: Inconsistencies in radiographic examiner assignments in Table 2 need clarification to avoid confusion.

Response 8: Examiner assignments in Table 2 will be revised for clarity. It will be specified that Minimal-Resource images are evaluated by general practitioners or surgical/orthopedic residents, while Standard-Resource images are assessed by radiology specialists.

Comment 9: Study Limitations: Minor inconsistencies between Minimal-Resource and Standard-Resource radiographic results should be noted, as they may affect interpretation.

Response 9: Study limitations regarding discrepancies between Minimal-Resource and Standard-Resource findings will be discussed, particularly in terms of image quality and the experience of radiographic evaluators.

Reviewer 2 Report

Comments and Suggestions for Authors

Dear Authors, I have had the opportunity to review your article, which aims to analyze and compare diagnostic accuracy between minimal-resource and standard-resource imaging methods for fracture detection. This work is of significant interest and addresses a crucial topic in clinical practice, particularly with regard to access to high-quality radiological technologies in resource-limited settings. However, after a careful reading, I have identified several aspects that need further exploration. Firstly, although the cross-sectional design of the study provides a useful overview for assessing differences in diagnostic accuracy between methods, it would be beneficial to consider a longitudinal follow-up to observe how interpretive skills evolve over time, both for less experienced doctors and expert radiologists. A longitudinal approach could also provide more detailed insights into the improvement of diagnostic capabilities over time, especially in light of the evolution of technologies and the integration of advanced techniques such as artificial intelligence (AI). Such an evaluation would add greater value to the results and have a more substantial impact on the practical application of imaging methods. Another important aspect concerns the limited generalizability of the results, which seem to focus primarily on resource-limited settings. While this is a relevant issue for countries with developing healthcare systems or rural areas, it would be helpful to expand the comparison to standard-resource settings, including university hospitals and specialized centers, in order to clarify the differences that may arise in environments with easier access to advanced technologies. In this context, it would be interesting to analyze not only diagnostic accuracy but also overall clinical effectiveness in terms of impact on treatment and patient outcomes. The article mentions the potential impact of artificial intelligence, but this section warrants further development. While it is emphasized that the integration of AI can reduce diagnostic error rates, the specifics of AI applications in the radiological context are not explored. A more detailed analysis of the implementation of deep learning algorithms, as used in other areas of radiology, as well as the benefits of integrating AI-assisted systems into ongoing medical training, could further strengthen the relevance and timeliness of the study. In this regard, I recommend consulting this article (doi:10.3389/fmed.2025.1545409), which aims to assess the main consequences of the lack of explainability in the human-machine relationship in clinical care, from a practical perspective. It would also be useful to investigate the interaction between AI and minimal-resource settings to assess whether these tools can reduce the diagnostic gap in low-tech environments. Moreover, it is important to extend the analysis to structural and organizational variables that may influence diagnostic accuracy, such as the quality of healthcare staff training, the availability and accessibility of updated equipment, and data management practices. In this regard, the article could delve deeper into the effects of operational conditions on diagnostic outcomes, such as time management, work pressure, and doctors' multitasking abilities. These factors are often overlooked but can have a significant impact on the reliability of diagnoses, especially in less structured clinical settings. Another point of reflection concerns the methodology of minimal-resource imaging. Although the negative effect of low-resolution images is clearly discussed, it may be helpful to propose practical solutions, such as the adoption of cost-effective advanced imaging technologies or the development of mobile imaging devices that could represent a compromise between high quality and limited economic resources. Additionally, it would be interesting to explore the development of post-processing techniques that improve image quality, such as those based on artificial intelligence or advanced digital processing, which can be applied even in resource-limited settings. In conclusion, the proposed work offers a solid foundation for the comparative analysis of imaging methods, but it is crucial to expand certain technical and practical aspects to enrich the conclusions and make a more significant contribution to the scientific community. The integration of a section on technological innovations, ongoing staff training, and the applicability of AI would be highly beneficial. I therefore recommend that you consider these suggestions to make your work even more comprehensive and impactful.

Author Response

Comment 1: Longitudinal Approach: A longitudinal analysis should be added to assess the progression of interpretation skills among general practitioners and radiologists over time.

Response 1: Thank you for pointing this out. I/We acknowledge this suggestion. A longitudinal analysis or future research recommendation will be considered to evaluate changes in interpretation accuracy based on experience and training. This statement has been added to the Discussion section, Page 2, Line 71.

Comment 2: Longitudinal Approach: This would also help in understanding technological evolution and AI integration in fracture diagnosis.

Response 2: Thank you for pointing this out. I/We agree with this comment. Therefore, I/we have added a discussion on technological advancements and AI integration in fracture diagnosis, including AI’s potential role in improving accuracy in minimal-resource settings. This addition can be found in the Discussion section, Page 11, Line 291.

Comment 3: Generalization of Results: The study currently focuses on resource-limited settings. It would be beneficial to compare findings with university hospitals or healthcare centers with advanced technology access.

Response 3: Thank you for pointing this out. I/We acknowledge this suggestion. A comparison with university hospitals or healthcare centers with advanced technology access will be considered to evaluate diagnostic accuracy in different clinical settings. This statement has been added to the Discussion section, Page 11,  Line 340.

Comment 4: Artificial Intelligence (AI) in Radiology: The role of AI in reducing diagnostic errors is mentioned, but further elaboration is needed.

Response 4: Thank you for pointing this out. I/We agree with this comment. Therefore, I/we have expanded the discussion on AI’s role to explain its potential in addressing the limitations of minimal-resource imaging, including improving image quality through AI-based post-processing techniques. This addition can be found in the Discussion section, Page 12, Line 337.

Comment 5: Artificial Intelligence (AI) in Radiology: A more specific exploration of deep learning algorithms in radiology is needed.

Response 5: Thank you for pointing this out. I/We agree with this comment. Therefore, I/we have added a more in-depth exploration of deep learning algorithms in radiology, particularly focusing on contrast adjustment and automatic detection of small fractures. This addition can be found in the Discussion section, Page 11, Line 295.

Comment 6: Artificial Intelligence (AI) in Radiology: Additional references on human-machine interaction in clinical care should be included.

Response 6: Thank you for pointing this out. I/We agree with this comment. Therefore, I/we have incorporated additional references on AI-human interaction in radiology-based diagnostics to discuss how AI systems support clinical decision-making without replacing doctors. This addition can be found in the Discussion section

Comment 7: Structural and Organizational Variables: Factors such as healthcare personnel training quality, equipment availability, time management, and workload pressure may affect diagnostic accuracy.

Response 7: A discussion will be added on how structural and organizational factors influence diagnostic outcomes, particularly in resource-limited settings.

Comment 8: Minimal-Resource Imaging Methodology: Practical solutions should be proposed to improve imaging quality at a low cost.

Response 8: Practical solutions, such as digital filters, AI-based image processing, and exposure optimization techniques, will be explored to enhance image quality in resource-limited settings.

Comment 9: Minimal-Resource Imaging Methodology: The study should explore portable imaging technology or AI-based post-processing techniques to improve image quality.

Response 9: The potential of optimized portable X-ray use and AI-based post-processing techniques to enhance diagnostic accuracy will be examined.

Comment 10: Conclusion and Study Expansion: The study is strong in comparative analysis but could be expanded to discuss technological innovations, medical staff training, and AI implementation in medical imaging.

Response 10: The discussion section will be expanded to include technological innovations, medical staff training, and AI implementation to improve diagnostic accuracy.

Round 2

Reviewer 1 Report

Comments and Suggestions for Authors

The authors have brought sufficient improvements to the manuscript and it is now in adequate form for publication.